# Variation in Wool Characteristics across the Body in a Herd of Alpacas Kept in Poland

**DOI:** 10.3390/ani11102939

**Published:** 2021-10-11

**Authors:** Aurelia Radzik-Rant, Małgorzata Wielechowska, Witold Rant

**Affiliations:** Department of Animal Breeding, Faculty of Animal Sciences, Warsaw University of Life Science-SGGW, Ciszewskiego 8 St., 02-786 Warsaw, Poland; malgorzata.czasza@gmail.com (M.W.); witold_rant@sggw.edu.pl (W.R.)

**Keywords:** alpaca, wool, fiber diameter, medullation, body site

## Abstract

**Simple Summary:**

Alpaca wool is luxurious and, hence, arouses great interest among consumers. However, the drawbacks of this wool are its variation in thickness and the proportion of medullated fibers. Knowing about variations in the quality characteristics of the wool on an animal’s body can help in properly evaluating and using this wool. This study has shown that the wool from the belly, front and hind legs, and the neck need to be separated from the total fleece. The most appropriate site from where to collect a representative sample of wool from the fleece for testing purposes is at the midside of the animal.

**Abstract:**

Wool characteristics vary depending on where on the body the wool is produced. Knowledge of this variation is important in order to separate the fleece into more homogenous parts. Similar parts from different animals can then be pooled to create batches of wool with similar characteristics. This will allow for better alpaca products with less variation. The aim of this study was to determine the variation in fiber diameter, medullation, and staple length across the body of alpacas from one herd. Wool samples were collected from 16 adult (3–5 years old) females: from the saddle (S), midside (MS), neck (N), and pieces (P). The mean fiber diameter (MFD) and medullation were measured using a projection microscope according to the IWTO-8-2011 standard. The fiber diameter of the pieces was greater (*p* < 0.05) than for the S, MS, and N areas. The highest medullation was found on the neck. The neck was also characterized by the shortest wool. The MFD for the fleece, excluding P, showed the strongest correlation (r = 0.927) with the MFD of MS. The study showed that due to the variation of fiber diameter, the incidence of medullation, and staple length, the wool from the pieces and the neck should be separated from the total fleece. The most appropriate site from where to collect a representative sample of wool from the fleece for testing purposes is the midside site of the animal.

## 1. Introduction

Fibers obtained from alpacas, which belong to the camelid family, are considered to be one of the most luxurious. Hence, these animals, farmed mainly in South American countries, have been successfully introduced into Australia, New Zealand, and other countries across various continents [1,2,3]. For many years, an interest in breeding these animals has also been noted in Poland [4]. Now that alpacas have been included as livestock, the development of this breed can significantly accelerate. In most South American countries, alpacas are kept primarily for fibers, but their meat is also used [5]. In other countries, including Poland, the most important resource from these animals is wool. With the development of the alpaca breed outside the South American continent, there is a growing need for research in the field of wool quality in the herds of these animals that are kept in different environmental conditions. There is very little research on the quality of Polish alpaca wool [6,7].

The quality of the fibers obtained from alpacas is primarily determined by the mean fiber diameter (MFD) and the presence of medullas. The staple length and crimping are also important. These parameters depend on many genetic and phenotypic factors, such as age, sex, and the color of the coat [2,3,5,7,8,9,10,11]. The MFD, not only of alpaca wool but also of sheep wool, mohair, cashmere, and vicuna wool, varies depending on the site of the body it is sampled from. This is indicated by studies completed on the variability of fiber diameter, staple length, and crimping of merino wool [12], Angora goats [13,14,15], alpacas [16,17], and vicunas [18].

Knox and Lamb [19] reported that fiber diameter varies significantly across the body. The coarsest fibers are on the belly and legs, while the side and neck are finer, and the finest is found on the back [20]. According to the study of Aylan-Parker and McGregor [16], the MFD on the back is about 1.5 µm less than from the side of the animal. In turn, McGregor et al. [17] found that the wool on the shoulder is the finest across 24 sites across the body.

In addition to the variation in fiber diameter for the different sites of the body, there can also be variations in the degree of medullation. According to Aylan-Parker and McGregor [16], the difference between the incidence of medullated fibers on the legs and on the saddle was approximately 11%. The variation in medullation within the fleece in Angora goats was also indicated in studies by Taddeo et al. [13].

In addition to the variation in fiber diameter across the body, the incidence of medullated fibers also varies across the body. These two factors affect the prickliness of the finished product [21,22,23]. Hence, breeding fine-wool alpacas with non-medullated fibers should be the focus of all breeders. This requires a measurement of fiber diameter and incidence of medullation that is representative of the whole fleece on individual animals [24,25,26,27].

The study of wool parameters in relation to the body site of the animal is important both for proper processing and the possibility of indicating an appropriate site for evaluation during both breeding and sales. Separating the parts of the whole fleece that have different values before processing, which is the standard practice in the sheep wool industry as well as in mohair production, can be helpful in obtaining textiles with the required functional properties.

Due to the scarcity of information about the quality and variability of fibers in the fleece of alpacas kept in Poland, the aim of this study was to determine the differences in MFD, medullation characteristics, and staple length across the body of animals, as well as to identify the optimum site from where to collect a wool sample for the testing of wool quality, based on one herd of Huacaya alpaca breed.

## 2. Materials and Methods

The study was conducted on 16 adult females of Huacaya alpacas from a herd located in the province of Silesia in Poland. The animals were the progeny of imported alpacas from Chile. The animals grazed natural pastures in spring, summer and autumn, and were kept in a barn during the winter, where they were fed a standard diet consisting of hay and concentrate with free access to water and mineral mixture. The shearing took place every 12 months in May using electric clippers by lying-down method. The average time of shearing was 20–30 min.

According to Polish law and the relevant EU directive, the experiment did not require approval from the local Ethical Committee because it was carried out on a private farm under production conditions [28].

The study was carried out on 16 females at the age of 3–5 years, born in Poland, which were not pregnant at the time of wool sampling. The body weight of tested animals ranged from 64–72 kg. The wool of sampled alpacas belonged to light color varieties: from white, light beige, beige, to light brown. The following sites of the body were taken into account in the wool quality tests: the saddle (S), which includes shoulder, central part and back of the animal; the midside (MS); the neck (N); and the pieces (P) that consist of the wool from the belly, and front and hind legs. Wool samples were collected during shearing from the left side of the animal. The midside was marked behind the tenth rib, halfway between the back line and the belly line [16], and one sample was taken for each animal. For the evaluation of the wool on other parts of the body (S, N, and P), after shearing and the fleece being laid out, wool samples were taken randomly from ten places for each body site. These samples, thoroughly blended into one sample, represented the tested body site. A total of 64 samples, weighing about 20 g, were collected, sealed in plastic bags, and stored for later analysis in the laboratory.

### 2.1. Measurements

Staple length of the samples was measure using a ruler with an accuracy of 1 mm. Wool samples were taken from each site after aqueous cleaning and drying and were placed under normal climate conditions for 24 h (65 ± 5% RH and 20 ± 2 °C) and were subjected to fiber diameter and medullation assessment using a projection microscope according to the IWTO-8-2011 standard [29]. From each well-mixed and parallelized sample, 1 mm snippets were cut from the middle of the sample. A minimum of 600 fibers was analyzed in each of the 64 samples, resulting in 38,400 fibers in total. The choice of the measurement method was dictated by the requirement to determine the type of fiber and type of medulla. Each fiber was classified according to its category of medulla into non-medullated, discontinuous medullated, or continuous medullated fibers.

The mean fiber diameter for all measured fibers in the sample (MFD), fiber diameter standard deviation (FD SD), and fiber diameter coefficient of variation (FD CV) were determined for total fleece (as average from all sampling sites), analyzed body sites, and for fleece where the data for pieces was excluded (fleece without pieces). The MFD, the FD SD, and the FD CV were also determined separately for each category of fibers.

The incidence of medullation was determined as a percentage share of all medullated fibers, regardless of the medulla type according to the formula (number of medullated fibers counted/total number of fibers measured)*100. Similarly, the percentage share of the individual fiber categories for the tested sites of the body was also determined.

### 2.2. Statistical Analyses

The statistical analysis of the data was performed using the one-way ANOVA, where the effect of sampling site was included in the model (df = 3). Moreover, the effect of MS, total fleece, and fleece without pieces have been determined. Tukey’s test was used to inspect all differences among the main factors [30].

*P* values < 0.05 were considered statistically significant. The results are presented as least-squares means (LSM) for each trait and standard deviation (±SD).

The Pearson correlation analysis among fleece without pieces and MS, S, and N for wool quality features was also undertaken.

Histograms and graphs were generated, showing the diameter distribution of category fiber within diameter classes.

## 3. Results

The MFD for all tested alpacas, regardless of the body site, was 22.0 µm; the FD SD and the FD CV were 5.1 µm and 22.4%, respectively. The total share of medullated fibers within the wool of the tested alpacas was 35.0%, the highest percentage of which were discontinuous medullated fibers. The total staple length was 12.8 cm (Table 1).

An analysis of the MFD for the designated sites of the body of the tested alpacas showed that the lowest MFDs were found on the saddle, larger in the midside, and the largest were found on the pieces. The MFD of the wool on the pieces was greater (*p*
*<* 0.05) than on the neck, on the saddle, and in the midside (Table 2). Similarly, wool from the pieces was characterized by a greater FD SD compared to wool from the neck and from the saddle (*p* < 0.05). Regarding the FD CV, no statistically significant differences were recorded between the examined body sites (Table 2).

It was also noted that there was no difference between the neck and the pieces for the incidence of medullated fibers. On the other hand, both the neck and the pieces were characterized by a much higher (*p* < 0.05) share of medullated fibers compared to the remaining analyzed sites of the fleece (Table 2). The saddle was characterized by the lowest degree of fiber medullation. The proportion of discontinuous medullated and continuous medullated fibers on the saddle was lower (*p* < 0.05) than on both the neck and pieces. The MS also showed a smaller (*p* < 0.05) proportion of this category of fibers compared to the neck and the pieces (Table 2).

Measurements of the staple length showed that the neck area was characterized by the shortest wool, while the longest wool was found in the midside. The wool on N was shorter compared to the wool on the midside (*p* < 0.05). The wool from the pieces was shorter (*p* < 0.05) than MS but slightly longer than the neck wool (Table 2).

The MFD measurements for all categories of fibers showed that the coarsest non-medullated fibers were found on the pieces. The diameter of these fibers was higher (*p* < 0.05) when compared to the remaining sites of the fleece (Table 3).

The FD SD value for non-medullated fibers on the MS, S, and N ranged from 3.5 to 3.6 µm; only the FD SD value for the pieces was higher, but statistically differed only with the neck (*p* < 0.05). Similar to non-medullated fibers, the discontinuous medullated fibers on the pieces were coarser in relation to this category of fibers on the saddle and on the neck (*p* < 0.05). In terms of the FD SD value, these types of fibers on the P showed higher values compared to the N, MS, and S (*p* < 0.05). Differences (*p* < 0.05) in the FD CV range for the discontinuous medullated fiber category were noted between the neck and the pieces. The continuous medullated fibers were characterized by a similar MFD for all analyzed sites of the alpaca fleeces. Only between N and P were there differences (*p* < 0.05) in the FD CV for this fiber category (Table 3).

A detailed analysis of the distribution of the proportion of fibers belonging to different categories of thickness classes confirmed there was a greater proportion of finer fibers on the saddle and in the midside, including having the largest proportion of non-medullated fibers compared to the neck and pieces (Figure 1, Figure 2, Figure 3 and Figure 4). On the N and P, apart from a smaller proportion of fibers with a diameter below 30 μm, the presence of a greater number of medullated fibers, especially with continuous medullas, was observed (Figure 3 and Figure 4). The fibers below 30 μm on the saddle were almost non-medullated, and only a small proportion of medullated fibers, mainly with discontinuous medullas, have been found (Figure 2).

Comparing the values of MFD and medullation characteristics obtained in the midside to the values representing the total fleece and the fleece after excluding pieces, it was shown that the values of MFD and FD SD were very similar (Table 4). The incidence of medullated fibers for the midside was lower in relation to the total fleece and the fleece without P, but this parameter did not show statistically significant differences, except for the proportion of fibers with continuous medullas, which on the MS was lower (*p* < 0.05) only in relation to the entire fleece (Table 4).

The relationship between the properties of MFD, medullation characteristics, and staple length defined for the fleece after excluding the pieces and the other sites of the body, turned out to be very strong (*p* < 0.05). The correlation coefficients ranged from 0.709 to 0.978. The MFD of the wool in the fleece without the P showed the strongest relationship with the MFD of the MS (0.927). A similarly strong relationship between the MS and the fleece without P was found in the MFD CV (0.964) and the incidence of medullated fibers (0.943). The staple length for the fleece without P showed the highest correlation with the saddle (Table 5).

## 4. Discussion

The MFD of the Huacaya alpacas’ wool in the studied herd was 21.97 µm, and the incidence of medullated fibers was 34.95%. Radzik-Rant and Wiercińska [7], in the tests of alpaca wool carried out in another herd in Poland, obtained MFD and incidence of medullated fibers of 25.3 μm and 68.9%, respectively. The wool thickness of alpacas tested by Valbonesi et al. [31] in Peru was 27.4 µm. In another study of the wool of Peruvian alpacas conducted by Pineres et al. [27], the mean fiber diameter and incidence of medullated fibers were 22.1 µm and 67.4%, respectively. McGregor and Butler [1] and McGregor [8], using various methods of measuring fiber diameter in alpacas in Australia, obtained the value of this parameter ranging from 28.8 μm to 29.9 μm. MFD in the studies of alpacas kept in the United States conducted by Lupton et al. [3] was 27.9 μm (Table 6).

A similar value of average staple length as in the present study (12.6 cm vs. 12.8 cm) in Peruvian alpacas was given by Frank et al. [5] in a review article. Wuliji [2] and McGregor [8,32], in studies on the wool of New Zealand and Australian alpacas, obtained a value of 9.9 cm, < 7.5–>15 and 9.4–7.7 cm for the average staple length, respectively.

The analysis of the examined wool characteristics for different sites of the body in this study showed that the samples representing multiple places on the saddle were more than 1 µm finer than the sample from the midside; it was also characterized by a lower FD CV and a lower incidence of medullated fibers (Table 2). Aylan-Parker and McGregor [16], in determining sampling sites similar to those in the present study, indicated that the finest fibers and the lowest variation in fiber diameter in their sample were collected from the midside. The MS was also characterized by the lowest proportion of medullated fibers. The proportion of the coarsest and largest incidence of medullated fibers was determined by these authors to be in the pieces. Similar results were obtained in the present study. The highest average fiber diameter, and an almost 45% share of medullated fibers, in the tested alpaca herd, was found in the pieces, but an even greater incidence of medullated fibers, compared to P, have been found on the neck. In turn, in studies by McGregor et al. [17], in determining the quality characteristics of the wool from 24 sites on the fleece, it was shown that the finest fibers covered the hind leg, then the shoulder, and the coarsest wool was found on the belly and front leg. In a study of the quality of vicuna wool by Quispe et al. [18], the differences in the MFD for different sites of the body did not exceed 1.5 µm. However, the finest fibers were found on the saddle in the anterior, central, and back sites closer to the dorsal line. Differences in the quality of mohair on the body of Angora goats were noted by Taddeo et al. [13] when analyzing the thickness and medullation of fibers from the belly, neck, midrib, britches, back, and shoulder. The finest wool was found on the back and britches, coarser on the shoulder and belly, and the coarsest was on the neck. In turn, McGregor [33] determined that the finest fibers were on the neck and shoulder, and the coarsest on the britches and back in American bison kept in Australia (Table 6). Sheep fleece was also characterized by variations in the thickness of the wool over the animals’ bodies. According to Fish et al. [12], changes in the value of this parameter run along the dorso-ventral line and from the back to the front.

In the analyzed herd of alpacas, the longest wool was found on the midside and on the saddle, while the shortest, as in the studies by McGregor et al. [17], was found on the shoulder (Figure 3). Changes in staple length for the different body areas were also recorded in studies on other species of animals. In vicuña, as in alpacas, the shortest wool was found around the neck, and longer wool was found at the back of the saddle [18]. In turn, in Angora goats and in American bison, the fibers on the neck were the longest [13,33] (Table 6).

The present research shows that the pieces should be separated from the fleece in order to make more effective use of the alpaca wool during the technological process. The samples of wool taken from the belly and front and hind legs were more than 4.5 µm coarser than the wool on the midside and more than 5.7 µm coarser than the wool on the saddle (Table 2). All fiber categories had the highest MFD on the pieces. Moreover, a large proportion of fibers above 30 µm, indicating a high prickling factor (Figure 2), leaves no doubt that this area needs to be removed from the entire fleece. According to McGregor et al. [17], in both Australia and Peru, the fleece of alpacas is usually divided into four components, i.e., the saddle, the neck, the skirting or pieces, and contaminated fibers (stained, vegetable matter). The present research, as well as other studies [17,18], indicate that in addition to the pieces, the wool from the neck should be separated from the entire fleece in view of its length and relatively high proportion of medullated fibers.

The differentiation of the qualitative characteristics of the wool, depending on the site of the body it comes from, hinders proper assessment. Designating a sampling site representative of the entire fleece is extremely important both in conducting breeding work and in assessing the raw material before it is processed. Alyan-Parker and McGregor [16], on the basis of tests conducted on wool thickness and incidence of medullated fibers in alpaca fleeces for sampling sites similar to those in this study, indicated the midside as a recommended site for the analysis of fiber diameter. However, to assess the variability of thickness and medullation, according to the authors, a sample taken from several sites on the saddle should also be taken into account. In the present study, the site that could best represent the MFD after excluding the pieces seems to be the midside. The MFD for MS was the same as the MFD of the fleece without P (20.8 µm vs. 20.8 µm), while the highest correlation coefficient was determined between the MFD of the entire fleece without the P and MS (0.927) (Table 4 and Table 5). The strongest relationship between MS and the fleece without P was also found in the variability of this feature. In the assessment of the incidence of medullated fibers, the proportion of fibers below 30 µm, and staple length, the recommended site could be samples taken either from MS or S (Table 4 and Table 5). McGregor et al. [17] indicated that the middle of the back saddle was the most appropriate site for determining the diameter of fibers in the fleece. Taddeo et al. [13], Fish et al. [12], and Person et al. [34] designated MS as being a representative site to sample for evaluating mohair from Angora goats and merino wool. In studies by Quispe et al. [18], the highest correlation with the thickness of the entire vicuna fleece was found on the central site of the saddle closer to the back line. However, samples taken from the back, central, and anterior of the saddle at different sites were correlated with the variation in thickness and staple length.

**Table 6 animals-11-02939-t006:** The results of studies on wool quality carried out on various animal species.

Species	Country	No. of Animals	No. of Sampling Sites	Mean Fiber Diameter (μm)	Fiber Diameter Coefficient of Variation (%)	Incidence of Medullated Fibers (%)	Staple Length (cm)	Reference
Alpaca	Poland	36	1	25.3	20.4	68.9	-	Radzik-Rant and Wiercińska [7]
Alpaca	Peru	36	1	22.1	-	67.4	-	Pinares et al. [27]
Alpaca	Australia	1100	1	24.0–29.9	-	20–50	7.5–15	McGregor [8]
Alpaca	Australia	836	1	28.8	25.4	-	-	McGregor and Butler [1]
Alpaca	USA	585	1	27.9	23.5	17.5	11.6	Lupton et al. [3]
Alpaca	Australia	120	4	27.5–37.6	24.3–30.6	24.4–44.5	-	Aylan-Parker and McGregor [16]
Alpaca	Peru	31	24	26.6–49.7	18–23.9	-	4.1–9.2	McGregor et al. [17]
Vicuna	Peru	30	12	13.6–14.3	19.3–20.7	-	2.3–3.3	Quispe et al. [18]
Angora goats	Argentina	40	6	31.2–37.5	20.4–21.8	2.3–3.7	11.4–12.4	Taddeo et al. [13]
American bison	Australia	16	5–7	17.6–19.4	24.0–27.0	-	2.7–4.7	McGregor [33]

## 5. Conclusions

The analysis of the mean fiber diameter, medullation characteristics, and staple length for different sites of the bodies of alpacas in the studied herd showed that there is a need to separate the pieces composed of wool from the belly and the front and hind legs from the total fleece.

It is also reasonable to exclude the wool from the neck and use it separately due to the high degree of medullation, which may cause an unpleasant prickling effect when using the finished products, and due to it having the shortest staple length of the fleece.

The site recommended for assessing tested wool characteristics could be the midside or a sample taken from several sites on the saddle.

## Figures and Tables

**Figure 1 animals-11-02939-f001:**
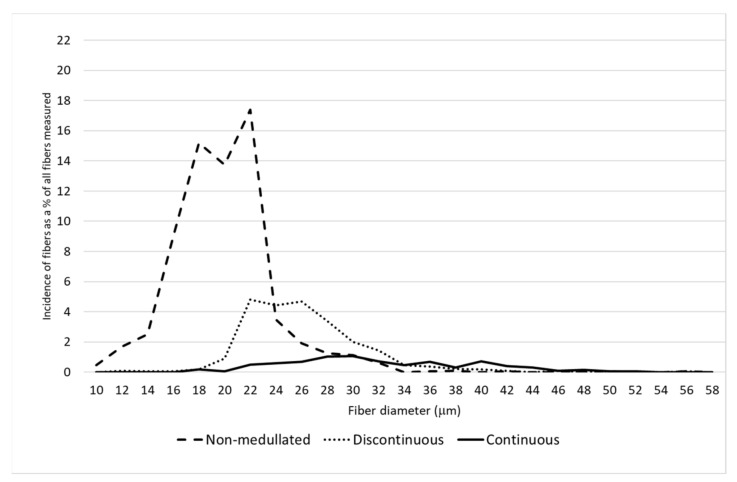
The distribution of each fiber category for midside.

**Figure 2 animals-11-02939-f002:**
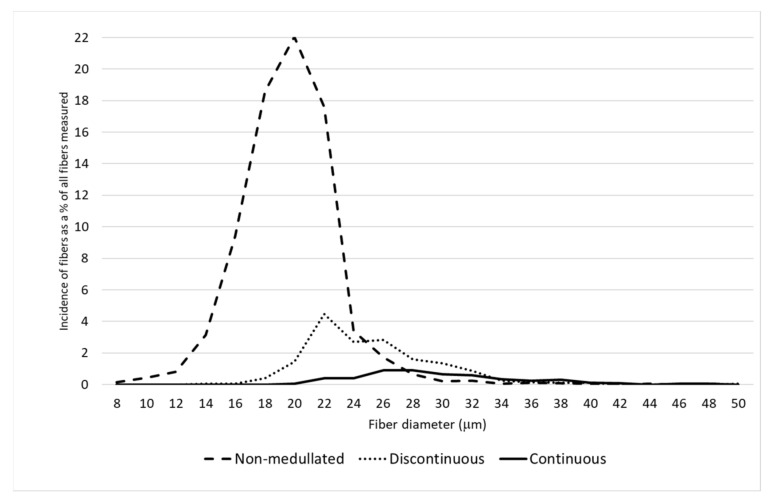
The distribution of each fiber category for the saddle.

**Figure 3 animals-11-02939-f003:**
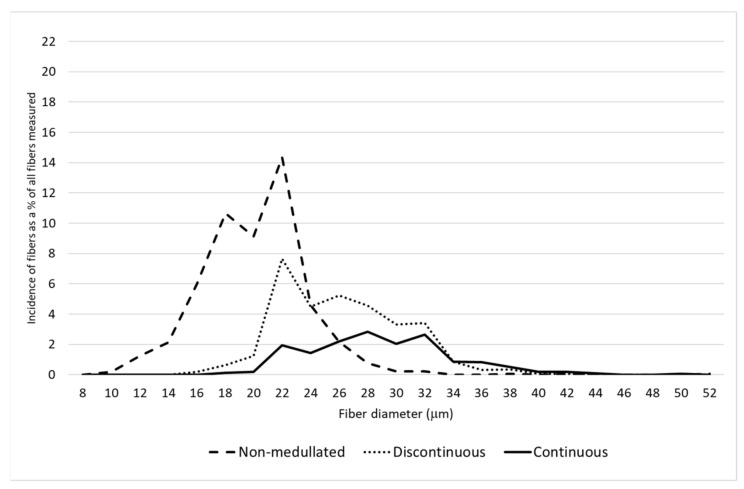
The distribution of each fiber category for the neck.

**Figure 4 animals-11-02939-f004:**
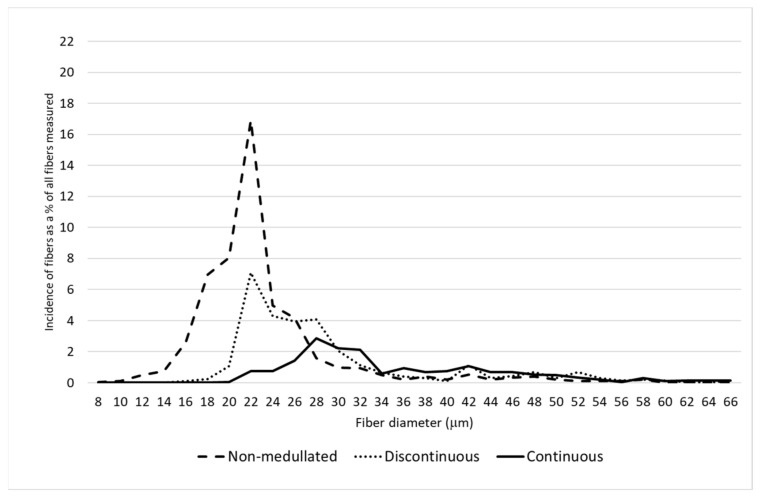
The distribution of each fiber category for the pieces.

**Table 1 animals-11-02939-t001:** The overall characteristics of wool quality for the alpacas being investigated.

Item	Total Fibers	Non-Medullated Fibers	Discontinuous Medullated Fibers	Continuous Medullated Fibers
	LSM ± SD	LSM ± SD	LSM ± SD	LSM ± SD
Mean fiber diameter (MFD) (μm)	22.0 ± 4.1	19.6 ± 3.1	24.3± 3.5	29.4 ± 4.0
Fiber diameter standard deviation (FD SD)	5.1 ± 2.1	3.9 ± 1.9	3.9 ± 2.2	4.7 ± 2.4
Fiber diameter coefficient of variation (FD CV) (%)	22.4 ± 4.6	19.2 ± 5.3	15.4 ± 5.4	15.7 ± 6.1
Fiber share (%)		65.1 ± 18.2	23.7 ± 14.1	11.3 ± 9.4
Incidence of medullated fibers (%)	35.0 ± 18.1			
Staple length (cm)	12.8 ± 3.5			

LSM: least-squares means, SD: standard deviation.

**Table 2 animals-11-02939-t002:** Wool quality attributes for the different body sites.

Item	Midside (MS)	Saddle (S)	Neck (N)	Pieces (P)	*p* Value
	LSM ± SD	LSM ± SD	LSM ± SD	LSM ± SD	
Mean fiber diameter (MFD) (μm)	20.8 ^a^ ± 2.4	19.7 ^a^ ±1.8	21.9 ^a^ ±2.3	25.5 ^b^ ± 5.9	0.000
Fiber diameter standard deviation (FD SD)	4.9 ± 1.3	4.1 ^a^ ± 0.6	4.7 ^a^ ± 0.56	6.4 ^b^ ± 3.6	0.005
Fiber diameter coefficient of variation (FD CV) (%)	23.3 ± 4.5	20.9 ± 2.4	21.7 ±1.3	23.8 ± 7.5	0.206
Non-medullated fibers (%)	72.5 ^a^ ± 14.8	78.7 ^a^ ± 11.9	53.9 ^b^ ± 15.9	55.1 ^b^ ± 16.4	0.000
Incidence of medullated fibers (%)	27.5 ^a^ ± 14.8	21.3 ^a^ ± 11.9	46.1 ^b^ ± 15.9	44.9 ^b^ ± 16.4	0.000
Discontinuous medullated fibers (%)	20.1 ± 12.8	15.6 ^a^ ± 10.6	29.1 ^b^ ± 16.0	30.0 ^b^ ± 11.9	0.003
Continuous medullated fibers (%)	7.5 ^a^ ± 3.5	5.7 ^a^ ± 2.6	17.0 ^b^ ± 9.8	15.0 ^b^ ± 12.4	0.000
Staple length (cm)	14.9 ^a^ ± 3.5	13.8 ± 3.5	10.9 ^b^ ± 3.0	11.7 ^b^ ± 2.8	0.001

LSM: least-squares means, SD: standard deviation. Values in the rows with the different letters differ significantly at: ^a^, ^b^: *p* < 0.5.

**Table 3 animals-11-02939-t003:** Wool quality characteristics for each fiber category of different body sites.

Item	Midside (MS)	Saddle (S)	Neck (N)	Pieces (P)	*p* Value
	LSM ± SD	LSM ± SD	LSM ± SD	LSM ± SD	
MFD of non-medullated fibers (μm)	19.0 ^a^ ± 1.7	18.0 ^a^ ± 0.9	18.9 ^a^ ± 1.4	22.6 ^b^ ± 4.6	0.000
FD SD of non-medullated fibers	3.6 ± 0.8	3.5 ± 1.3	3.5 ^a^ ± 0.6	5.0 ^b^ ± 3.2	0.040
FD CV of non-medullated fibers (%)	18.6 ± 3.0	19.1 ± 6.4	18.3 ± 2.5	21.0 ± 7.524.0	0.420
MFD of discontinuous medullated fibers (μm)	24.0 ± 1.8	23.2 ^a^ ± 1.3	23.5 ^a^ ± 2.5	26.6 ^b^ ± 5.6	0.009
FD SD of discontinuous fibers	3.6 ^a^ ± 1.2	3.4 ^a^ ± 0.9	3.2 ^a^ ± 0.8	5.5 ^b^ ± 3.8	0.006
FD CV of discontinuous medullated fibers (%)	14.8 ± 4.2	14.7 ± 3.4	13.6 ^a^ ± 2.1	18.5 ^b^ ± 8.6	0.038
MFD of continuous medullated fibers (μm)	29.4 ± 3.8	28.3 ± 2.5	28.6 ± 3.5	31.5 ± 5.1	0.062
FD SD of continuous medullated fibers	5.1 ± 1.6	4.2 ± 1.4	3.8 ^a^ ± 1.1	5.8 ^b^ ± 3.8	0.049
FD CV of continuous medullated fibers (%)	17.5 ± 4.9	15.1 ± 5.7	13.1 ± 2.7	17.2 ± 9.0	0.104

LSM: least-squares means, SD: standard deviation. Values in the rows with the different letters differ significantly at: ^a^, ^b^: *p* < 0.5.

**Table 4 animals-11-02939-t004:** Wool quality characteristics for the midside, total fleece, and fleece without pieces.

Item	Midside	Total Fleece	Fleece Without Pieces	*p* Value
	LSM ± SD	LSM ± SD	LSM ± SD	
Mean fiber diameter (MFD) (μm)	20.8 ± 2.4	22.0 ±2.9	20.8 ± 2.0	0.282
Fiber diameter standard deviation (FD SD)	4.9 ± 1.3	5.1 ± 1.4	4.6 ± 0.8	0.495
Fiber diameter coefficient of variation (FD CV) (%)	23.3 ± 4.5	22.4 ± 3.4	22.0 ± 2.5	0.532
Non-medullated fibers (%)	72.5 ± 14.8	65.1 ± 13.1	68.4 ± 13.4	0.275
Incidence of medullated fibers (%)	27.5 ± 14.8	35.0 ± 13.0	31.6 ± 13.4	0.275
Discontinuous medullated fibers (%)	20.1 ± 12.8	23.7 ± 12.1	21.6 ± 12.6	0.683
Continuous medullated fibers (%)	7.5 ^a^ ± 3.5	11.3 ^b^ ± 5.3	10.0 ± 4.7	0.047

LSM: least-squares means, SD: standard deviation. Values in the rows with the different letters differ significantly at: ^a^, ^b^: *p* < 0.5.

**Table 5 animals-11-02939-t005:** The correlations between body sites and the fleece without pieces for wool quality characteristics in the alpacas being investigated.

Item	Midside (MS)	Saddle (S)	Neck (N)
Mean fiber diameter (MFD) (μm)	0.927 *	0.901 *	0.914 *
Fiber diameter standard deviation (FD SD)	0.956 *	0.961 *	0.864 *
Fiber diameter coefficient of variation (FD CV) (%)	0.964 *	0.921 *	0.709 *
Non-medullated fibers (%)	0.943 *	0.939 *	0.949 *
Incidence of medullated fibers (%)	0.943 *	0.939 *	0.949 *
Discontinuous medullated fibers (%)	0.960 *	0.949 *	0.955 *
Continuous medullated fibers (%)	0.824 *	0.663 *	0.964 *
Staple length (cm)	0.973 *	0.978 *	0.935 *

*: *p* < 0.05.

## Data Availability

The data presented in this study are available on request from the corresponding author. The data are not publicly available to preserve the privacy of the data.

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
