# Peer review of "Variation in Wool Characteristics across the Body in a Herd of Alpacas Kept in Poland"

_animals, 2021, doi:10.3390/ani11102939_

Round 1
Reviewer 1 Report
Dear Authors,
thanks for following my suggestions. I found your manusrcript improved and suitable for publication in Animals journal.
Author Response
September, 09, 2021
Response to reviewer
Dear Reviewer,
Thank you very much for your in-depth review, which improved the quality of the manuscript.
Thank you also for accepting the changes that have been made according to your recommendations.
Kind regards
Reviewer 2 Report
Dear Authors,
I have started to review this paper but decided to stop because the paper needs a lot of work and basically needs to be rewritten before it is publishable. I understand that English is not the authors' first language and therefore I strongly advise them to get some help with the English language. If the paper was in a form that I could have edited directly, then I could have helped but it is difficult in its current format. The first two pages that I have reviewed indicates the amount of work that this paper requires.
Kind regards
Reviewer’s comments
Title: Variation in the selected wool characteristics on the different body sites in the heard of alpacas kept in Poland
Authors: Radzik-Rant A and Wielechowska M
This paper describes the variation across an alpaca fleece to determine how the fleece should be broken up to compile uniform batches of wool, and identifying the best place to take a wool sample to test fibre characteristics. It adds value to our understanding of variation across the fleece in Alpacas. However, I strongly advice the authors to read the classic paper “Measurement as an aid to selection in breeding sheep for wool production” by Helen Newton Turner that was published in June 1956 in Animal Breeding Abstracts Volume 24 (2) 87-118.
Title
I suggest the authors change the title to something like the following.
“Variation of wool characteristics across the body in Alpacas in Poland”
Line 9. Suggestion. “However, fibre thickness and the proportion of medullated fibres vary significantly across the body. Knowledge of this variation can help to characterise the different regions in order to class wool in more uniform batches suitable for processing. ”
Line 13. Suggestion. Change “.. could be the middle of the side,…” to “could be the midside centre, approximately xx centimeters from the centre line along the back, and xx centimeters from the shoulder of the animal.”
Lines 15 and 16. Change “these variations” to “this variation”
Line 16. Suggest authors change sentence starting with “ Knowledge…” with
“Knowledge of this variation is important to determine which sites across the body should be sampled to obtain the best indication of the fibre characteristics of a fleece. This will increase the fleece value and best use of the wool from different sites across the body “
Lines 17-27. Suggest to change section starting with “The aim of this study…” with
“The aim of this study was to quantify the variation of fibre diameter, fibre medullation and staple strength across the bodies of Alpacas. Wool samples were collected from the saddle (S), midside (MS), neck (N) and pieces (P) on 16 adult females from one herd. Mean fibre diameter (MFD) and medullation of the fibres were measured on each sample with a projection microscope according to the IWTI8-2011 standard. MFD of the pieces was greater (P<0.05) than for the S, MS and N areas. The neck had the shortest wool that also had the most medullated fibres. MFD of the fleece was strongest correlated (r=0.927) with MFD at the midside site of the fleece. This study shows that the differences in fibre diameter, incidence of medullated fibres and staple length across the body, indicate that the pieces and wool from the neck, should be separated from the fleece, and placed in separate categories to compile most homogenous batches for processing. The midside is also the most appropriate sampling site to obtain an indication of the fleece’s fibre characteristics.
Line 32. Suggestion to change sentence starting with “Hence, these animals farmed mainly ….” to “Hence, these animals, farmed mainly for wool and meat [5] in South American countries, have….”
Line 36. Suggest to delete sentence starting with “ In most South American countries……is also used [5]. “
Line 38. Replace “resource” with “product”
Lines 39-42. Suggest to delete sentences starting with “With the development…” and end with …of Polish alpaca wool [6,7].” This should be in the last paragraph of the introduction.
Line 43. Suggestion. Delete lines 63 to 68 and modify line 43 to read as follows. “The quality of the fibers obtained from Alpacas is primarily determine by the MFD and the presence of medullated fibres. These traits impact on the prickliness [21-23] and dyeability of the finished product. Therefore breeding finer fibres that are less medullated should be part of a breeding program for Alpacas [24-27].
Line 45. Change sentence starting with “These parameters…” to “These characteristics depend on genetic and environmental factors, such as feeding conditions, diet, age…….etc.
Line 46. Suggest you modify the sentences starting from “The mean fiber diameter, not only….” to the end of the paragraph with
“However, the MFD, staple length and number of crimps per unit length, also varies at different sites across the body in Alpacas [[16,17], similarly as in sheep [12], Angora goats [13-15] and vicunas [18]. Knox and Lamb [19]….”
Line 55. Replace “side” with “midside”
Line 56. Change sentence “ …. sampled 24 locations indicated…..” with “ ….sampled 24 different locations across the body and concluded that the finest wool in Alpaca is found on the shoulder.”
Suggestion. Delete the sentence on line 61 starting with “The variation in medullation within the fleece…..” and change the sentence on line 58 to the following.
“In addition to the variation in fiber diameter at the different site of the body, the degree of medullation can also differ across the body as shown in Angora goats [13].
I have decided to stop here because I think this paper needs to be rewritten completely. If it was in an acceptable Word format then it would have been easier but in its current state it is difficult to rewrite.
Author Response
September, 09, 2021
Response to reviewer
Dear Reviewer,
Thank you very much for your review, but due to the fact that the review was not completed in full, I cannot respond to the proposed changes.
The manuscript was previously evaluated by three reviewers, who had no objections to the English language. I would also like to mention that before submission the manuscript was proofread by the native speaker (Certificate of English Editing attached).
Moreover, in the comments of the Academic Editor it was stated that the manuscript has been substantially improved from the earlier version.
Kind regards

Round 2
Reviewer 2 Report
Dear Authors,
Below are my comments on the manuscript. It is very difficult to read and needs extensive improvement in style. I have made some suggestions up to line 136 but then decided that the paper needs to be totally rewritten and I can not do that for you. I hope the following suggestions will give you some clues, where the paper can be improved.
Here are my suggestions and questions
Suggested title: Variation in wool characteristics across the body of Alpacas in Poland
Line 13. Replace sentence starting with “The appropriate sites…” with “The most appropriate site from where to collect a representative sample of wool from the fleece for testing purposes, is at the midside site of the animal.
If you include “several points on the saddle” then you have to specify where across the saddle, and whether these different samples from the saddle should be pooled. That opens a whole range of problems as it means that the samples should be blended into one sample for testing because nobody will test multiple wool samples from across the saddle.
Line 15. Suggestion to replace first sentence. “Wool characteristics varies depending where on the body the wool is produced. “
Lines 15-17. Suggestion to replace first 2 sentences. “ Knowledge of this variation is important in order to separate the fleece into more homogenous parts. Similar parts from different animals can then be pooled into create batches of wool with similar characteristics. This will allow for better alpaca products with less variation.
Line 17. Suggestion. “The aim of this study was to determine the variation in fibre diameter, medullation and staple length across the body of Alpacas.”
Line 19. How old were these 16 females?
Line 20. Replace “middle side” with “midside” which is the standard name/term for this site in Merino sheep.
Line 20. Insert “mean” before “fibre diameter”
Line 22. Replace “for” with “of” so sentence reads “Fibre diameter of the pieces was greater…”
Line 25. Suggestion “The study showed that due to the variation of fibre diameter, incidence of medullation and staple length, the wool from the neck should be separated from that of the saddle. The most appropriate site from where to collect a representative sample of wool from the fleece for testing purposes, is at the midside site of the animal.
Line 29. Keywords. Delete “the” before “body site”
Line 53. Suggestion. Knox and Lam [19] reported that fiber diameter varies significantly across the body. The coarsest fibres are…”
Line line 56. Insert “the” before “study”
Line 56. Delete “on 4 body sites of Alpacas”
Line 58-59. Suggestion “ In turn, McGregor et al [17] found that the wool on the shoulder is the finest across 24 sites across the body.
Lines 65-84. Suggestion. “In addition to the variation in fibre diameter across the body, the incidence of medullated fibres also varies across the body. These two factors affect the prickliness of the finished product [21-23]. Hence breeding fine wool Alpacas with no non-medullated fibres should be a focus of all breeders. This requires a measurement of fiber diameter and the incidence of medullation that is representative of the whole fleece on individual animals. In Merino sheep it has been shown that the midside is the best representative site across the body (See Turner 1956; Measurement as an aid to selection. Animal Breeding Abstracts). As Alpacas are normally kept on pasture during summer and in barns during winter where they are fed a constant diet, this may impact on the variation of fibre quality traits across the fleece. The aim of this study was to quantify the variation of fibre traits across the fleece to identify the optimum site from where to collect a wool sample for the testing of wool quality of Alpacas in Poland.
Line 86. Undelete “16 adult female”
Line 87. Delete “The heard was …. Chile” . Replace with “The animals were the progeny of imported Alpacas from Chile.
Line 88-94. Suggestion. “The animals grazed natural pastures in spring, summer and autumn, and were kept in barns where they were fed a standard diet consisting of hay and concentrate with free access to water and mineral mixture. Shearing took place every 12 months in May.
Delete lines 98-101 up to before “ The following sites….”.
Line 102 -104. Suggestion “ … the saddle (S), which includes the shoulder, central part and back of the animal, the neck (N) and the pieces (P) that consist of the wool from the belly and front and hind legs as shown in Figure 1.
Line 106. Replace the sentence starting with “For the evaluation….” with “Samples were collected from different sites across the body as shown in Figure 1.
Was the midside sampled?
The figure shows 30 sites per animal, and it is not clear how you got to 72 samples per animal??
Line 114-123. Suggestion. Staple length of the staples were measure using a ruler with an accuracy of 1 mm (I hope this is what you’ve done). Wool samples were then washed, cleaned and then dried under natural conditions over 24 hours to a relative humidity (RH) of 65 ± 5% and to 20 ± 2 degrees Celsius. From each staple a 1mm snippet were cut from the middle of the staple. A minimum of 600 fibres were analysed in each of the 72 samples, resulting in 43,200 snippets of fibres in total. Fibre diameter and medullation were measured using a projection according to the IWTO-8-2011 standard [29]. Medullated fibres were classified as being non-medullated, discontinuously medullated or continuously medullated fibres. The incidence of different medullation types was determined as a percentage of the total number of fibres measured (number of medullated fibres counted/total number of fibres measured*100).
How many fibre snippets did you measure on each sample? Did you actually measured fibre diameter and medullation on all 43,200 snippets.??? How many staples were involved, 72 or 30 as shown on the figure?
Line 124 to 136. It is not clear for which part the means were calculated. If on the staples on each site, then the following sentence may do. “The mean fibre diameter (MFD), standard deviation of fibre diameter (SD) and coefficient of variation of fibre diameter were calculated for each staple”.
Author Response
October, 01, 2021
Response to reviewer
Dear Reviewer,
Thank you very much for your review.
- The title of the manuscript has been changed on: “Variation in Wool Characteristics Across the Body in the Herd of Alpacas kept in Poland”
- Line 13. The sentence has been changed as suggested by the reviewer.
- Line 15. The sentence has been replaced as suggested by the reviewer.
- Line 15-17. The sentence has been changed as suggested by the reviewer.
- Line 19. The females were at the age of 3 to 5 years.
- The term “middle of the side” has been replaced on “midside” all over the manuscript.
- Line 20 and 22. The changes were introduced as suggested by the reviewer.
- Line 25. The sentence has been changed as suggested by the reviewer.
- Line 29. Changed according to reviewer suggestion.
- Line 53. The sentence has been changed as suggested by the reviewer.
- Line 56. Changed according to reviewer suggestion.
- Lines 58-59. The sentence has been changed as suggested by the reviewer.
- Lines 65-84. The paragraph has been rewritten as suggested by the reviewer.
- Line 87. The sentence was replaced as suggested by the reviewer.
- Lines 88-94. The sentences have been changed.
- Lines 98-101. The sentences have not been deleted because they were added in accordance with previous referee suggestion.
- Lines 102-104. Changed according to reviewer suggestion.
- Line 106. Changed according to reviewer suggestion.
- Lines 114-123. The paragraph has been rewritten.
- The wool samples in this study were prepared as follow: From the midside one sample was taken of each animal (behind the tenth rib, halfway between the back line and the belly line). For the evaluation of the wool on other parts of the body (S, N and P), wool samples were taken randomly from ten places for each body site. These samples, were thoroughly blended into one sample for testing. Hence 4 wool samples was taken for each alpaca. In total 64 wool samples were analyzed.
The manuscript was previously evaluated by three reviewers, who had no objections to the English language. I would also like to mention that before submission the manuscript was proofread by the English-native speaker (Certificate of English Editing attached).
Kind regards

Round 3
Reviewer 2 Report
Dear Authors,
Thank you for improving the manuscript. It has improved considerably and is now easier to read.
Regards